# The Effects of Dietary Fermented Soybean Meal Supplementation on the Growth, Antioxidation, Immunity, and mTOR Signaling Pathway of Juvenile Coho Salmon (*Oncorhynchus kisutch*)

**Qin Zhang, Qiuyue Yang, Mengjie Guo, Fanghui Li, Meilan Qin, Yi Xie, Jian Xu, Yongqiang Liu \* and Tong Tong \***

Guangxi Marine Microbial Resources Industrialization Engineering Technology Research Center, Guangxi Key Laboratory for Polysaccharide Materials and Modifications, School of Marine Sciences and Biotechnology, Guangxi Minzu University, 158 University Road, Nanning 530008, China; qzhang1220@126.com (Q.Z.); m18860930655@163.com (Q.Y.); 17852021210@163.com (M.G.); lfhx272@163.com (F.L.); 15077030520@163.com (M.Q.); xy463317779@163.com (Y.X.); wingaboy@163.com (J.X.)
\* Correspondence: liuyongqiang110@126.com (Y.L.); ttong1028@126.com (T.T.)

**Abstract:** This experiment aims to investigate the effects of partial replacement of fish meal (FM) by soybean meal (SBM) and/or fermented soybean meal (FSBM) on the growth, serum biochemistry, digestion, antioxidation, immunity, and mTOR signaling pathway of juvenile coho salmon (*Oncorhynchus kisutch*). Four iso-nitrogen and iso-lipid diets were designed and fed to four groups of juvenile coho salmon (152.25 ± 2.96 g) in triplicate for 10 weeks. The four diets were the G0 diet (control group, containing 28% FM protein), the G1 diet (containing 10% SBM protein and 18% FM protein), the G2 diet (containing 5% SBM protein, 5% FSBM protein, and 18% FM protein), and the G3 diet (containing 10% FSBM protein and 18% FM protein). It was found that compared with the G0 diet (control group), the activities of catalase (CAT), superoxide dismutase (SOD), pepsin, trypsin, $\alpha$-amylase, and lipase, and the expression levels of mammalian target of rapamycin (*mtor*), protein kinase B (*akt*), phosphatidylinositol 3-kinase (*pi3k*), plant ribosome S6K protein kinase (*s6ks*), and lysozyme (*lyz*) genes, of juvenile coho salmon fed the G3 diet increased significantly ($p < 0.05$), and the expression levels of tumor necrosis factor (tnf-$\alpha$), interleukin-1$\beta$ (il-1$\beta$), and interleukin-6 (il-6) genes of juvenile coho salmon fed the G3 diet decreased significantly ($p < 0.05$). However, there were no significant differences in growth, muscle composition, and serum biochemistry of juvenile coho salmon fed the G3 diet compared with the G0 diet ($p > 0.05$). In conclusion, replacing 10% FM protein with FSBM protein could improve the digestion, antioxidation, immunity, and mTOR signaling pathway of juvenile coho salmon.

**Keywords:** coho salmon; fermented soybean meal; antioxidation; immune; mTOR signaling pathway

**Key Contribution:** Partial replacement of fish meal with fermented soybean meal could improve the expression levels of mTOR signaling pathway genes of juvenile coho salmon, which indicates that fermented soybean meal could promote the protein metabolism of juveniles.



## 1. Introduction

Fish meal (FM) is the main protein source of aquatic feed, which has the advantages of rich protein content, balanced amino acid structure, and high nutrient digestibility [1]. However, due to the rapid growth of the aquaculture industry in recent years, the market demand for FM is increasing, the prices of FM continue to rise, and the supply of FM is also significantly limited [2]. Soybean meal (SBM) has been widely used as a feed protein for many years. However, the large amount of anti-nutrient factors contained in SBM can reduce aquatic animals' growth performance and nutrient digestion [3]. Fermentation technology of SBM is the most effective method to degrade polymeric protein and anti-nutrient



factors to enhance the nutrition of SBM [4]. The advantages of fermented soybean meal (FSBM) as a substitute for FM have been widely confirmed by the studies. There are many published studies regarding the replacement of FM with FSBM with contradictory results. One study showed that using FSBM instead of FM to feed spotted seabass (*Lateolabrax maculatus*) has high growth performance and immune ability, and the optimum replacement level of FM with the FSBM was estimated to be in the range of 26.9–37.1% [5]. However, up to 30% FM in the diets of juvenile yellowfin (*Acanthopagrus latus*) could be replaced by FSBM, and the growth performance and whole-body proximate compositions of fish were not affected by dietary FSBM level [6]. Therefore, considering the different response strategies of different species to FSBM and little research on the effect of FSBM instead of FM on salmon, finding suitable levels of FSBM and probiotics is very important for the development of the salmon industry.

The mammalian target of rapamycin (mTOR) protein signaling pathway has been widely demonstrated to be an important pathway for protein synthesis [7], which regulates cell growth, cell cycle, and other physiological functions mainly through the phosphatidylinositol 3-kinase/protein kinase B/mammalian target of rapamycin (PI3K/AKT/mTOR) pathway [8]. Studies have shown that the expression levels of mTOR-signaling-pathway-related genes were related to dietary protein content, and high dietary protein and lipid contents could promote the expression levels of mTOR-signaling-pathway-related genes [9,10]. In addition, amino acid levels, antistrophic factors, and small molecule active substances in the diet also had regulatory effects on mTOR activity [11–13].

Coho salmon (*Oncorhynchus kisutch*) is one of the promising new Pacific salmon species in China due to its rich nutrition, delicious meat, and high economic value [14]. FM is considered by far the most advantageous protein source for coho salmon. However, there are not many reports on coho salmon fed FSBM instead of FM. The substitution of FM for FSBM was studied in coho salmon to integrate an assessment of its effects. In this experiment, the SBM fermented by a high protease-producing strain to replace part of FM was used to feed juvenile coho salmon to explore its effects on growth, serum biochemistry, digestion, antioxidation, immunity, and mTOR signaling pathway. This experiment provides a scientific reference for the study of substituting FM with FSBM and the optimization and development of commercial environmental feedstuffs for coho salmon.

## 2. Materials and Methods

### 2.1. Experimental Diet

A high protease-producing strain S7 (*Enterococcus lactis*) was screened from mangrove soil (21°81′66″ N, 108°58′46″ E, Qinzhou, China). The optimal fermentation conditions were determined in the early stage, and the main preparation steps of FSBM were as follows: First, the mass ratio of SBM to distilled water was 1:1. Second, 8% *Enterococcus* lactate S7 was inoculated into SBM (the volume of liquid medium was 8% of the mass of SBM, the bacteria content was about $8 \times 10^8$ CFU/mL). Third, it was fermented at 37 °C for 36 h. Fourth, FSBM was dried at 30 °C air stream for 12 h, and the moisture of FSBM was less than 10%. Fifth, the dried FSBM was packed in the sample bag and stored at −20 °C for use. The nutritional composition and anti-nutritional factor content of soybean meal and fermented soybean meal were determined by Shanghai Panrui Technology Co., LTD (Shanghai, China), as shown in Table 1.

Referring to previous studies [15,16], the replacement amount of FM protein in this experiment was determined to be 10%. According to the nutritional requirements of coho salmon, four iso-nitrogen (crude protein was 42.93%) and iso-lipid (crude fat was 15.21%) diets were formulated. The control group (G0) contained 28% FM protein, and in the G1, G2, and G3 groups, 10% of FM protein was replaced by SBM and/or FSBM protein. The G1 group was replaced with 10% SBM protein; the G2 group was replaced with 5% SBM protein and 5% FSBM protein; and the G3 group was replaced with 10% FSBM protein, as shown in Table 2.

**Table 1.** Nutritional composition and anti-nutritional factor content of soybean meal and fermented soybean meal.

| Index | Soybean Meal | Fermented Soybean Meal |
|---|---|---|
| Crude protein (%) | 46.81 ± 0.21 | 55.21 ± 0.33 |
| Crude lipid (%) | 1.84 ± 0.05 | 1.93 ± 0.09 |
| Crude ash (%) | 8.62 ± 0.22 | 7.45 ± 0.19 |
| Moisture (%) | 9.35 ± 0.21 | 9.64 ± 0.15 |
| Polypeptide (%) | 1.39 ± 0.06 | 21.83 ± 0.21 |
| Trypsin inhibitors (mg/g) | 66.13 ± 1.58 | 11.35 ± 0.95 |
| Glycinin (mg/g) | 141.13 ± 3.65 | 24.15 ± 1.75 |
| β-Conglycinin (mg/g) | 105.01 ± 2.74 | 26.39 ± 1.16 |
| Urease (U/g) | 8.01 ± 0.18 | 1.09 ± 0.05 |
| pH | 7.15 ± 0.02 | 6.44 ± 0.04 |

**Table 2.** Experimental diet formula (g/100 g of dried feed) and approximate composition (%, dry matter percentage).

| Ingredients | G0 | G1 | G2 | G3 |
|---|---|---|---|---|
| Fish meal | 40.07 | 25.99 | 25.99 | 25.99 |
| Soybean meal | 0.00 | 21.20 | 10.76 | 0.00 |
| Fermented soybean meal | 0.00 | 0.00 | 7.95 | 15.89 |
| Poultry meal | 10.02 | 10.00 | 10.00 | 10.00 |
| Shrimp meal | 10.02 | 10.00 | 10.00 | 10.00 |
| Wheat middling | 17.88 | 17.84 | 17.84 | 17.84 |
| Starch | 3.02 | 3.01 | 3.01 | 3.01 |
| Cellulose | 8.53 | 0.00 | 2.67 | 5.31 |
| Fish oil | 4.00 | 5.53 | 5.53 | 5.53 |
| Soybean oil | 4.00 | 4.00 | 4.00 | 4.00 |
| Calcium dihydrogen phosphate | 1.01 | 1.01 | 1.01 | 1.01 |
| Mineral premix [1] | 0.52 | 0.52 | 0.52 | 0.52 |
| Vitamin premix [2] | 0.52 | 0.52 | 0.52 | 0.52 |
| Choline | 0.30 | 0.30 | 0.30 | 0.30 |
| Vitamin C | 0.10 | 0.10 | 0.10 | 0.10 |
| Approximate composition | | | | |
| Fish meal protein | 28.00 | 18.00 | 18.00 | 18.00 |
| Poultry meal protein | 6.51 | 6.51 | 6.51 | 6.51 |
| Shrimp meal protein | 6.01 | 6.01 | 6.01 | 6.01 |
| Soybean meal protein | 0.00 | 10.00 | 5.00 | 0.00 |
| Fermented soybean meal protein | 0.00 | 0.00 | 5.00 | 10.00 |
| Flour protein | 2.41 | 2.41 | 2.41 | 2.41 |
| Crude protein | 42.93 | 42.93 | 42.93 | 42.93 |
| Fish meal fat | 4.94 | 3.41 | 3.41 | 3.41 |
| Poultry meal fat | 1.22 | 1.22 | 1.22 | 1.22 |
| Shrimp meal fat | 1.03 | 1.03 | 1.03 | 1.03 |
| Fish oil fat | 4.00 | 5.53 | 5.53 | 5.53 |
| Soybean oil fat | 4.00 | 4.00 | 4.00 | 4.00 |
| Crude fat | 15.21 | 15.21 | 15.21 | 15.21 |
| Ash (%) | 7.26 | 7.64 | 7.54 | 7.33 |
| Moisture (%) | 9.52 | 9.57 | 9.53 | 9.49 |

Note: [1] The mineral premix provides the following per kg diet: $AlK(SO_4)_2 \cdot 12H_2O$ 123.70 mg; $CaCl_2$ 17,879.80 mg; $CuSO_4 \cdot 5H_2O$ 31.70 mg; $CoCl_2 \cdot 6H_2O$ 48.90 mg; $FeSO_4 \cdot 7H_2O$ 707.40 mg; $MgSO_4 \cdot 7H_2O$ 4316.80 mg; $MnSO_4 \cdot 4H_2O$ 31.10 mg; $ZnSO_4 \cdot 7H_2O$ 176.70 mg; KCl 1191.90 mg; KI 5.30 mg; NaCl 4934.50 mg; $Na_2SeO_3 \cdot H_2O$ 3.40 mg; $Ca(H_2PO_4)_2 \cdot H_2O$ 12,457.00 mg; $KH_2PO_4$ 9930.20 mg. [2] The vitamin premix provides following per kg diet: retinal palmitate, 10,000 IU; cholecalciferol, 4000 IU; α-tocopherol, 75 IU; menadione, 22.00 g; thiamine HCl, 40.00 g; riboflavin, 30.00 g; D-calcium pantothenate, 150.00 g; pyridoxine HCl, 20.00 g; meso-inositol, 500.00 g; D-biotin, 1.00 g; folic acid, 15.00 g; ascorbic acid, 200.00 g; niacin, 300.00 g; cyanocobalamin, 0.30 g.

All the raw materials (purchased from Shandong Kangkerun Co., LTD., Weifang, China) were ground into 80 mesh of fine powder with a hammer mill and then mixed in a roller mixer for 15 min. Distilled water (40% mass) was added to make a hard dough. A single screw extruder was used to obtain floating pellets with a diameter of $2.0 \times 3.0$ mm. Then, the feed particles were dried in the airflow at 30 °C for 12 h, and the moisture of the feed particles was less than 10%. The dried feed particles were packed in the sample bag and stored at $-20$ °C for use.

### 2.2. Experimental Fish and Acclimatization

Five hundred juvenile coho salmon were purchased from a rainbow trout breeding farm in Benxi City, China. After disinfection with potassium permanganate solution with a concentration of 1/100,000–1/50,000, they were domesticated in the outdoor breeding system of a rainbow trout breeding farm (Benxi, China) for 2 weeks. The use of juvenile coho salmon in this experiment was approved by the Ethics Committee of Guangxi Minzu University (No: GXMZU 2021-006).

During domestication, water temperature was 10–18 °C, dissolved oxygen was $\geq 6.0$ mg/L, water intake was $\geq 100$ L/s, water surface velocity was $\geq 2$ cm/s, pH was 7.5–7.8, and natural light was used. A third of the water was changed every day, and the basic feed was fed three times a day (8:00, 12:00, and 16:00). The feeding condition and activity of juvenile coho salmon was observed every day, and the feeding amount was adjusted according to the feeding condition to achieve satiety.

After being acclimatized for 2 weeks, a total of 240 juvenile coho salmon ($152.25 \pm 2.96$ g) were randomly divided into 4 groups in triplicate, resulting in 12 net cages with 20 fish per net cage ($1 \times 1 \times 0.5$ m$^3$). As described in acclimatization progress, juvenile coho salmon were cultured in the same aquaculture system, and the juveniles were fed with one of the four diets (above Table 2) for 10 weeks.

### 2.3. Sampling

After the culture experiment for 10 weeks, the juvenile coho salmon were starved for 24 h. Six juvenile coho salmon were randomly selected from each net cage and anesthetized with 40 mg/L 3-aminobenzoic acid ethyl ester methane sultanate (MS-222, ADAMAS reagent, China). The body weight and body length of each juvenile coho salmon were measured. Then, the blood of juvenile coho salmon was collected from the caudal vein. The blood samples were placed at 4 °C for 24 h, centrifuged at 4 °C and 4000 g for 15 min, and the upper serum was taken into a new 1.5 mL centrifuge tube. The dorsal major muscle sample was collected into a 20 mL centrifuge tube. The liver sample was weighed and collected into a 20 mL centrifuge tube. The intestine sample was collected into a 20 mL centrifuge tube. All the above samples were stored at $-80$ °C.

### 2.4. Growth Performance

The calculation formulas of weight gain rate (WGR), specific growth rate (SGR), feed conversion ratio (FCR), protein efficiency rate (PER), hepatosomatic index (HSI), condition factor (CF), and survival rate (SR) are as follows:

$$\text{WGR (\%)} = \frac{\text{Final body weight (g)} - \text{Initial body weight (g)}}{\text{Initial body weight (g)}} \times 100$$

$$\text{SGR (\%/day)} = \frac{[\ln(\text{Final body weight) (g)} - \ln(\text{Initial body weight) (g)}]}{\text{days}} \times 100$$

$$\text{FCR} = \frac{\text{Total feed intake (g)}}{\text{Final body weight (g)} - \text{Initial body weight (g)}}$$

$$\text{PER (\%)} = \frac{\text{Final body weight (g)} - \text{Initial body weight (g)}}{\text{Total intake of crude protein weight (g)}} \times 100$$

$$\text{CF (\%)} = \frac{\text{Body weight (g)}}{\left[\text{Body length (cm)}\right]^3} \times 100$$

$$\text{SR (g)} = \frac{\text{Final amount of fish}}{\text{Inital amount of fish}} \times 100$$

### 2.5. Determination of Muscle Composition

Muscle moisture was determined by the oven drying constant weight method at 105 °C (GB/T 6435-2014, Chinese standards). Muscle crude protein was determined by the Kjeldahl nitrogen determination method (GB/T 6432-2018, Chinese standards). Muscle crude fat was determined by the Soxhlet extraction method (GB/T 6433-2006, Chinese standards). Muscle ash was determined by the burning method in a muffle furnace at 550 °C (GB/T 6438-2007, Chinese standards).

### 2.6. Determination of Biochemical Indexes and Enzymes

The glucose (GLU), total cholesterol (T-CHO), total protein (TP), albumin (ALB), alkaline phosphatase (AKP), alanine aminotransferase (GPT), and glutamic oxalic amino-transferase (GOT) in serum; the total antioxidant capacity (T-AOC), superoxide dismutase (SOD), catalase (CAT), and malondialdehyde (MDA) in the liver; and the pepsin, trypsin, alpha-amylase, and lipase in the intestine were measured using an enzyme label analyzer (TECAN, Infinite F50, Männedorf, Zurich, Switzerland) and assay kits, respectively, according to the kits' instruction manuals. All the assay kits were produced by Nanjing Jiancheng Bioengineering Institute (Nanjing, China), and all the instruction manuals can be found and downloaded at http://www.njjcbio.com/ (accessed on 1 July 2023).

### 2.7. Determination of Gene Expression

The expression of mammalian target of rapamycin (*mtor*), plant ribosome S6K protein kinase (*s6ks*), protein kinase B (*akt*), phosphatidylinositol 3-kinase (*pi3k*), lysozyme (*lyz*), tumor necrosis factor α (*tnf-α*), interleukin 1β (*il-1β*), and interleukin 6 (*il-6*) genes in the liver were determined by real-time quantitative polymerase chain reaction (RT-qPCR). *β-actin* was selected as the internal reference gene. Based on the mRNA sequences of coho salmon published in the National Center for Biotechnology Information (NCBI), Primer Premier 7.0 (Premier Biosoft International, Palo Alto, Canada) was used to design the forward and reverse primers. The primers were synthesized by Shanghai Sangon Bioengineering Technology Service Co., LTD. (Shanghai, China), as shown in Table 3.

The RT-qPCR method of Zhang et al. was applied [17], and the brief steps were as follows: First, the total RNA was extracted from the liver (about 100 mg) using the RNAiso Plus Kit produced by Biotech Ltd. Co., Takara, Dalian, China. The RNA quantity and purity were determined by spectrophotometry and assessed based on the absorbance ratio of 260:280 nm using an ND-2000 spectrophotometer (Thermo, Waltham, MA, USA). The RNA integrity was evaluated based on gel electrophoresis in a 1% (*w/v*) agarose TAE gel stained with Gel RedTM nucleic acid stain (UVP, Upland, CA, USA). Second, 1 μg of total RNA was reverse transcribed into cDNA using the RT Master Mix for qPCR (AG) produced by Biotech Ltd. Co., Takara, Dalian, China. The reverse transcription processes were 30 °C for 10 min, 60 °C for 30 min, 95 °C for 5 min, 5 °C for 5 min, and 1 cycle. Third, RT-qPCR assays were performed using an RT-qPCR Detection System (LightCycler 96, Roche, Basel, Switzerland) and the SYBR Green Pro Taq HS Premix (AG) produced by Biotech Ltd. Co., Takara, Dalian, China. The RT-qPCR reactions were carried out with 12.5 μL of SYBR Premix Ex Taq, 1 μL of primer F, 1 μL of primer R, 2 μL of cDNA, and 8.5 μL of dH$_2$O, in a total reaction volume of 25 μL. The RT-qPCR thermal cycling conditions were preheated, 95 °C for 10 s, 1 cycle, followed by 40 cycles of three steps: denaturation, 95 °C for 60 s;

annealing, 60 °C for 30 s; and extension, 72 °C for 90 s. The reaction specificity was detected by the 95 °C melting curve.

**Table 3.** Primer sequence of genes for RT-qPCR.

| Primer Name | Primer Sequence | Product Size (bp) | TM (°C) | GenBank |
|---|---|---|---|---|
| *β-actin* [1] | F: CCAAAGCCAACAGGGAGAA <br> R: AGGGACAACACTGCCTGGAT | 91 | 60.0 | XM_031822094.1 |
| *Mtor* [2] | F: CTTCGCCAACTACCTCCG <br> R: TGCCCTCTTCACCTCAAACT | 139 | 60.0 | XM_020506200.2 |
| *Akt* [3] | F: GCAGCCATCCTACAAATC <br> R: TGAAACAGGGTCCACAAG | 178 | 60.0 | XM_031831237.1 |
| *pi3k* [4] | F: CCAGTGGCTCAAGGACAAGAACAG <br> R: GGATGAAGGTGGCTACGCAGTATC | 98 | 60.0 | XM_020466892.2 |
| *s6ks* [5] | F: CAGCACCTGAGCAGCAGTTAGC <br> R: CTCGGATCGGCAGTGGAAAGTTC | 131 | 60.0 | XM_020465833.2 |
| *Lyz* [6] | F: GCTGTTGTTGTTCTCCTGCTTGTG <br> R: TGTTTCCAGCGTAGCCATCCATTC | 109 | 60.0 | XM_020457770.2 |
| *tnf-α* [7] | F: GGCGAGCATACCACTCCTCT <br> R: TCGGACTCAGCATCACCGTA | 125 | 60.0 | XM_020497470.2 |
| *il-1β* [8] | F: GCGACATGGTGCGTTTCCTTTT <br> R: TGTCTACCGGTTTGGTGTAGTCCT | 129 | 60.0 | XM_020475860.2 |
| *il-6* [9] | F: GAGCTACGTAACTTCCTGGTTGAC <br> R: GCAAGTTTCTACTCCAGGCCTGAT | 134 | 60.0 | XM_020472300.2 |

Note: [1] *β-actin*: reference gene. [2] *mtor*: mechanistic target of rapamycin. [3] *akt*: serine/Threonine kinase, or protein kinase B (PKB). [4] *pi3k*: phosphatidylinositide 3-kinases. [5] *s6ks*: ribosomal protein S6 kinase. [6] *lyz*: lysozyme. [7] *tnf-α*: tumor necrosis factor α. [8] *il-1β*: interleukin 1β. [9] *il-6*: interleukin 6.

*2.8. Data Statistics*

All the data were preliminarily processed by Microsoft Excel 2021 (Microsoft Corporation, Washington, WA, USA), and a one-way analysis of variance (ANOVA) was performed by IBM SPSS 21 (International Business Machines Corporation, Armonk, NY, USA). The normality and homogeneity of variances among groups were tested. Tukey's multiple range test was used to compare each other. When $p < 0.05$, it is significantly different. GraphPad Prism9 (GraphPad software) was used for graphing, and the least significant difference method was used for the significance test. Results were expressed as Mean $\pm$ standard deviation (Mean $\pm$ SD). The relative expression levels of genes were calculated by the $2^{-\Delta\Delta CT}$ method [18].

**3. Results**

*3.1. Growth Performance*

Compared with the G0 (control group), the final weight, WGR, SGR, CF, and PER of the juvenile coho salmon fed the G1 and G2 diets were significantly decreased ($p < 0.05$), and the HSI and FCR of the juvenile coho salmon fed the G1 and G2 diets were significantly increased ($p < 0.05$). However, there were no significant differences in the final weight, WGR, SGR, HSI, CF, FCR, and PER of the juvenile coho salmon fed the G3 and G0 diets ($p > 0.05$), and there was no significant difference in the SR of the juvenile coho salmon fed the dietary partial replacement of FM by SBM and/or FSBM ($p > 0.05$), as shown in Table 4.

*3.2. Muscle Composition*

Compared with the G0 (control group), the content of crude protein of the juvenile coho salmon fed the G1 and G2 diets was significantly decreased ($p < 0.05$), and the content of crude protein of the juvenile coho salmon fed the G3 diet was significantly increased ($p < 0.05$). However, there were no significant differences in the moisture, crude fat, and ash of the juvenile coho salmon fed the dietary partial replacement of FM by SBM and/or FSBM ($p > 0.05$), as shown in Table 5.

**Table 4.** Effect of fermented soybean meal on growth performance of juvenile coho salmon.

| Index | G0 | G1 | G2 | G3 | F-Values | *p*-Values |
|---|---|---|---|---|---|---|
| Initial weight | 152.25 ± 2.96 | 152.25 ± 2.96 | 152.25 ± 2.96 | 152.25 ± 2.96 | 0.000 | 1.000 |
| Final weight | 512.35 ± 10.20 [a] | 416.26 ± 9.84 [c] | 472.18 ± 10.61 [b] | 524.28 ± 8.92 [a] | 72.549 | 0.000 |
| WGR [1] (%) | 336.52 ± 6.70 [a] | 273.41 ± 6.46 [c] | 310.13 ± 6.97 [b] | 344.35 ± 5.86 [a] | 72.552 | 0.000 |
| SGR [2] (%/day) | 1.73 ± 0.03 [a] | 1.44 ± 0.04 [c] | 1.62 ± 0.04 [b] | 1.77 ± 0.03 [a] | 66.083 | 0.000 |
| HIS [3] (%) | 1.42 ± 0.02 [c] | 1.50 ± 0.02 [a] | 1.46 ± 0.02 [b] | 1.44 ± 0.02 [bc] | 13.176 | 0.002 |
| CF [4] (%) | 1.24 ± 0.03 [a] | 1.07 ± 0.04 [b] | 1.11 ± 0.03 [b] | 1.22 ± 0.02 [a] | 23.229 | 0.000 |
| FCR [5] | 1.53 ± 0.05 [c] | 2.09 ± 0.08 [a] | 1.72 ± 0.06 [b] | 1.48 ± 0.04 [c] | 71.489 | 0.000 |
| PER [6] (%) | 2.92 ± 0.07 [a] | 2.13 ± 0.06 [c] | 2.56 ± 0.012 [b] | 2.97 ± 0.07 [a] | 66.551 | 0.000 |
| SR [7] (%) | 93.33 ± 2.89 | 91.67 ± 2.89 | 93.33 ± 2.89 | 95.00 ± 0.00 | 0.889 | 0.487 |

Notes: All above data are mean ± SD (*n* = 3). Different superscript letters in the same row indicate significant differences among the data (*p* < 0.05). [1] WGR: weight growth rate. [2] SGR: specific growth rate. [3] HSI: hepatosomatic index. [4] CF: condition factor. [5] FCR: feed conversion ratio. [6] PER: protein efficiency ratio. [7] SR: survival rate.

**Table 5.** Effect of fermented soybean meal on muscle composition of juvenile coho salmon.

| Index | G0 | G1 | G2 | G3 | F-Values | *p*-Values |
|---|---|---|---|---|---|---|
| Moisture (%) | 72.17 ± 0.16 | 72.79 ± 0.88 | 71.74 ± 0.30 | 72.48 ± 0.74 | 1.678 | 0.248 |
| Crude protein (%) | 20.73 ± 0.24 [b] | 19.67 ± 0.25 [c] | 20.10 ± 0.33 [c] | 21.22 ± 0.21 [a] | 18.181 | 0.001 |
| Crude fat (%) | 5.73 ± 0.22 | 5.59 ± 0.12 | 5.68 ± 0.14 | 5.87 ± 0.18 | 1.431 | 0.304 |
| Ash (%) | 1.48 ± 0.04 | 1.53 ± 0.04 | 1.51 ± 0.04 | 1.46 ± 0.05 | 1.589 | 0.267 |

Notes: All above data are mean ± SD (*n* = 3). Different superscript letters in the same row indicate significant differences among the data (*p* < 0.05).

### 3.3. Serum Biochemical Indexes

Compared with the G0 (control group), the GLU, T-CHO, TP, ALB, and AKP of the juvenile coho salmon fed the G1 and G2 diets were significantly decreased (*p* < 0.05). However, there were no significant differences in the GLU, T-CHO, TP, ALB, and AKP of the juvenile coho salmon fed the G3 and G0 diets (*p* > 0.05), and there were no significant differences in the GOT and GPT of the juvenile coho salmon fed the dietary partial replacement of FM by SBM and/or FSBM (*p* > 0.05), as shown in Table 6.

**Table 6.** Effect of fermented soybean meal on serum biochemical indexes of juvenile coho salmon.

| Index | G0 | G1 | G2 | G3 | F-Values | *p*-Values |
|---|---|---|---|---|---|---|
| GLU [1] (mmol/L) | 6.60 ± 0.28 [a] | 4.20 ± 0.10 [c] | 4.75 ± 0.26 [b] | 6.20 ± 0.18 [a] | 83.585 | 0.000 |
| T-CHO [2] (mmol/L) | 8.02 ± 0.25 [a] | 3.62 ± 0.22 [c] | 6.79 ± 0.23 [b] | 8.15 ± 0.28 [a] | 220.104 | 0.000 |
| TP [3] (µg/mL) | 62.37 ± 6.03 [a] | 40.07 ± 2.30 [b] | 44.00 ± 4.81 [b] | 63.60 ± 5.29 [a] | 94.329 | 0.000 |
| ALB [4] (µg/mL) | 31.12 ± 1.07 [a] | 18.15 ± 2.94 [b] | 18.35 ± 2.44 [b] | 33.81 ± 3.05 [a] | 32.854 | 0.000 |
| AKP [5] (U/L) | 184.71 ± 13.23 [a] | 73.69 ± 5.50 [c] | 124.66 ± 16.32 [b] | 183.21 ± 10.07 [a] | 377.06 | 0.000 |
| GOT [6] (U/L) | 3.93 ± 0.32 | 3.43 ± 0.35 | 3.55 ± 0.29 | 3.63 ± 0.37 | 1.223 | 0.363 |
| GPT [7] (U/L) | 3.63 ± 0.29 | 3.60 ± 0.40 | 3.99 ± 0.40 | 3.37 ± 0.53 | 1.151 | 0.387 |

Notes: All above data are mean ± SD (*n* = 3). Different superscript letters in the same row indicate significant differences among the data (*p* < 0.05). [1] GLU: glucose. [2] T-CHO: total cholesterol. [3] TP: total protein. [4] ALB: albumin. [5] AKP: alkaline phosphatase. [6] GOT: glutamic oxalic aminotransferase. [7] GPT: alanine aminotransferase.

### 3.4. Antioxidant Capacity

Compared with the G0 (control group), the content of MDA of the juvenile coho salmon fed the G1 diet and the activities of CAT and SOD of the juvenile coho salmon fed the G3 diet were significantly increased (*p* < 0.05). The content of T-AOC and the activity of CAT and SOD of the juvenile coho salmon fed the G1 and G2 diets were significantly decreased (*p* < 0.05). However, there was no significant difference in the content of T-AOC of the juvenile coho salmon fed the G3 and G0 diets (*p* > 0.05), and there was no significant difference in the content of MDA the juvenile coho salmon fed the G2, G3, and G0 diets (*p* > 0.05), as shown in Table 7.

**Table 7.** Effect of fermented soybean meal on antioxidant capacity of juvenile coho salmon.

| Index | G0 | G1 | G2 | G3 | F-Values | *p*-Values |
|---|---|---|---|---|---|---|
| T-AOC [1] (mmol/gprot) | 0.70 ± 0.03 [a] | 0.16 ± 0.01 [b] | 0.18 ± 0.02 [b] | 0.72 ± 0.07 [a] | 212.364 | 0.000 |
| CAT [2] (U/mgprot) | 103.22 ± 15.27 [b] | 42.46 ± 2.37 [d] | 68.82 ± 3.74 [c] | 311.69 ± 23.33 [a] | 305.071 | 0.000 |
| SOD [3] (U/mgprot) | 266.22 ± 23.26 [b] | 93.26 ± 2.47 [d] | 174.91 ± 14.97 [c] | 387.75 ± 32.57 [a] | 104.646 | 0.000 |
| MDA [4] (nmol/mgprot) | 2.09 ± 0.23 [b] | 3.60 ± 0.09 [a] | 2.21 ± 0.14 [b] | 2.07 ± 0.08 [b] | 75.718 | 0.000 |

Notes: All above data are mean ± SD (*n* = 3). Different superscript letters in the same row indicate significant differences among the data ($p < 0.05$). [1] T-AOC: total antioxidant capacity. [2] CAT: catalase. [3] SOD: superoxide dismutase. [4] MDA: malondialdehyde.

### 3.5. Digestion Capacity

Compared with the G0 (control group), the activities of the pepsin, trypsin, α-amylase, and lipase of the juvenile coho salmon fed the G3 diet were significantly increased ($p < 0.05$), and the activities of the pepsin, trypsin, α-amylase, and lipase of the juvenile coho salmon fed the G1 diet, and the activities of the pepsin and trypsin of the juvenile coho salmon fed the G2 diet were significantly decreased ($p < 0.05$). However, there were no significant differences in the α-amylase and lipase of the juvenile coho salmon fed the G2 and G0 diets ($p > 0.05$), as shown in Table 8.

**Table 8.** Effect of fermented soybean meal on digestion capacity of juvenile coho salmon.

| Index | G0 | G1 | G2 | G3 | F-Values | *p*-Values |
|---|---|---|---|---|---|---|
| Pepsin (U/mgprot) | 12.02 ± 0.63 [b] | 4.18 ± 0.40 [d] | 8.62 ± 0.82 [c] | 17.32 ± 1.25 [a] | 124.414 | 0.000 |
| Trypsin (U/mgprot) | 2268.31 ± 117.36 [b] | 1028.04 ± 117.85 [d] | 1566.38 ± 134.81 [c] | 2586.07 ± 111.89 [a] | 100.913 | 0.000 |
| α-amylase (U/mgprot) | 0.40 ± 0.04 [b] | 0.31 ± 0.02 [c] | 0.41 ± 0.03 [b] | 0.61 ± 0.07 [a] | 24.654 | 0.000 |
| Lipase (U/mgprot) | 26.18 ± 1.10 [b] | 24.71 ± 0.50 [c] | 26.88 ± 0.69 [b] | 32.24 ± 1.50 [a] | 30.939 | 0.000 |

Notes: All above data are mean ± SD (*n* = 3). Different superscript letters in the same row indicate significant differences among the data ($p < 0.05$).

### 3.6. Gene Expression

Compared with the G0 (control group), the expression of the *mtor*, *akt*, *pi3k*, and *s6ks* genes of the juvenile coho salmon fed the G3 diet were significantly increased ($p < 0.05$), and the expression of the *mtor* and *s6ks* genes of the juvenile coho salmon fed the G1 and G2 diets, and the expression of the *akt* and *pi3k* genes of the juvenile coho salmon fed the G1 diet were significantly decreased ($p < 0.05$). However, there were no significant differences in the expression of the *akt* and *pi3k* genes of the juvenile coho salmon fed the G2 and G0 diets ($p > 0.05$), as shown in Figure 1.

Compared with the G0 (control group), the expression of the *tnf-α*, *il-1β*, and *il-6* genes of the juvenile coho salmon fed the G1 and G2 diets and the expression of the *lyz* gene of the juvenile coho salmon fed the G3 diet were significantly increased ($p < 0.05$), and the expression of the *tnf-α*, *il-1β*, and *il-6* genes of the juvenile coho salmon fed the G3 diet and the expression of the *lyz* gene of the juvenile coho salmon fed the G1 diet were significantly increased ($p < 0.05$). However, there was no significant difference in the expression of the *lyz* gene of the juvenile coho salmon fed the G2 and G0 diets ($p > 0.05$), as shown in Figure 2.

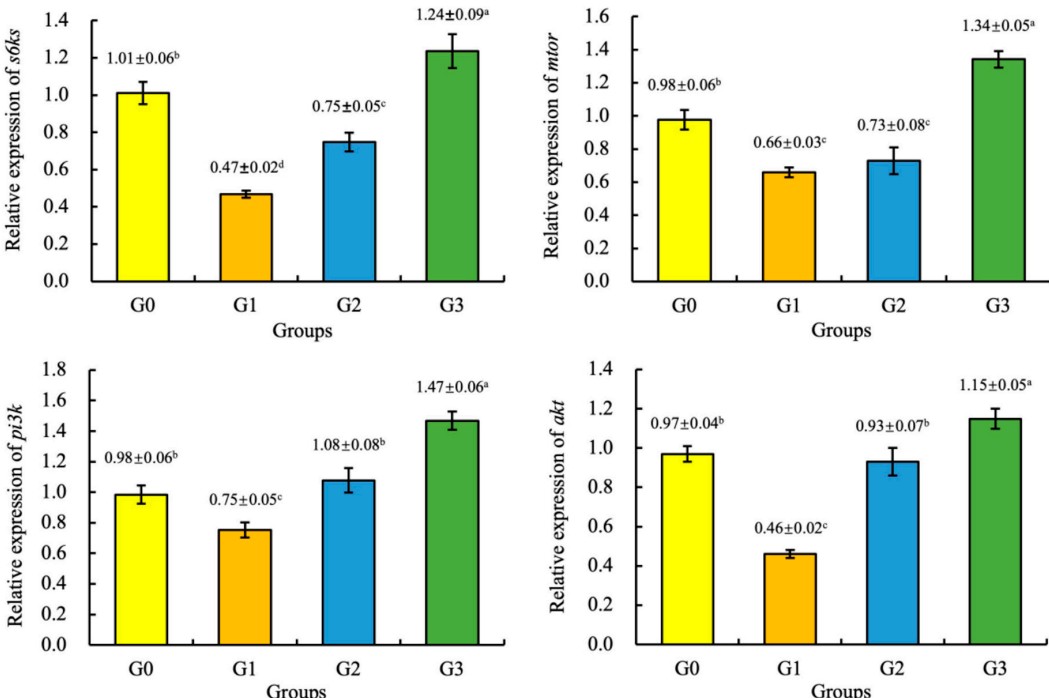

**Figure 1.** Effect of fermented soybean meal on the relative expression of the ribosomal protein S6 kinase 1 (*s6ks*), mechanistic target of rapamycin (*mtor*), serine/threonine kinase (*akt*), and phosphatidylinositide 3-kinases (*pi3k*) genes in the liver of juvenile coho salmon. All the above data are mean $\pm$ SD ($n = 3$). Different superscript letters in the same row indicate significant differences among the data ($p < 0.05$).

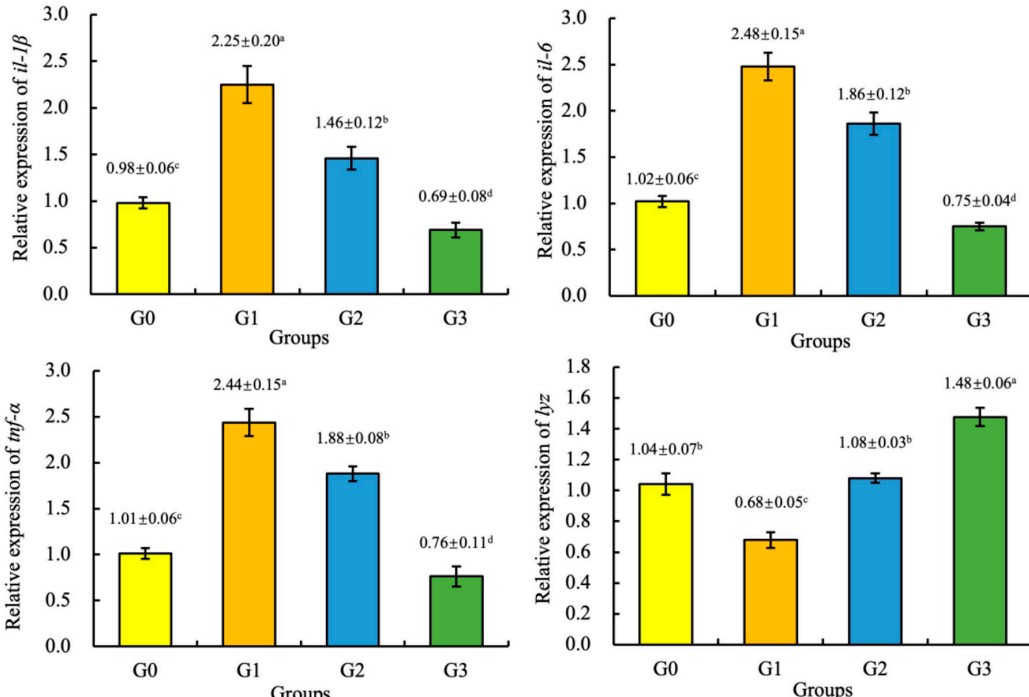

**Figure 2.** Effect of fermented soybean meal on the relative expression of the interleukin 1β (*il-1β*), interleukin 6 (*il-6*), tumor necrosis factor α (*tnf-α*), and lysozyme (*lyz*) genes in the liver of juvenile coho salmon. All the above data are mean $\pm$ SD ($n = 3$). Different superscript letters in the same row indicate significant differences among the data ($p < 0.05$).

## 4. Discussion

In this study, we found that replacing part of FM with SBM significantly decreased the growth performance of juvenile coho salmon, and FSBM had no significant effect on the growth performance of juvenile coho salmon, but there was an upward trend. The reasons are supposed to be the following: First, SBM lacked some minerals and essential amino acids and contained many anti-nutrient factors, which could affect the digestion and utilization of SBM protein by fish [19]. However, the FSBM could optimize the content of amino acids and reduce the content of anti-nutrient factors, which could reduce the negative effect of SBM on experimental animals [20]. Second, proteins with small molecular masses were more easily absorbed and utilized by the experimental animals than those with large molecular masses [21]. The strain in this study had high protease activity, which could decompose large molecular proteins in SBM into small molecular peptides and amino acids. Thus, the absorption and utilization rate of SBM protein could be improved, and the growth performance of fish could be promoted [22]. Third, SBM would produce some bioactive peptides and some unknown nutritional factors during fermentation, which could improve intestinal health and digestive enzyme activity and then promote the growth performance and development of aquatic animals [23,24]. Sørensen et al. found that SBM protein could induce enteritis, which was not conducive to the growth performance and development of Atlantic salmon (*Salmo salar*) [25]. Li et al. found that the growth performance of Nile tilapia (*Oreochromis niloticus*) was significantly improved by feeding FSBM [26]. Han et al. found that feeding fermented cottonseed meal could improve the growth performance of white shrimp (*Litopenaeus vannamei*) [27].

In this study, we found that partial replacement of FM with SBM significantly decreased the expression of the *mtor*, *akt*, *pi3k*, and *s6ks* genes of juvenile coho salmon, and partial replacement of FM with FSBM significantly increased the expression of the *mtor*, *akt*, *pi3k*, and *s6ks* genes of juvenile coho salmon. The reasons are supposed to be that SBM contains many anti-nutritional factors, which leads to the inability of feed protein to be digested and utilized by fish, thus reducing the availability of feed protein. Reduced availability of feed protein inhibited the expression of genes related to the mTOR signaling pathway, thereby preventing the deposition of fish protein [28]. In addition, during the fermentation of SBM, the fermentation strain not only reduced the content of anti-nutrient factors but also degraded macromolecular proteins and released many amino acids and active small peptides [29]. Both amino acids and active small peptides were the activators of the mTOR signaling pathway and promoted the expression of related genes [30–32]. Similar studies have shown that Wacyk et al. substituted the plant protein raw materials for FM in rainbow trout (*Oncorhynchus mykiss*) feed and found that the expression of the related genes of the mTOR signaling pathway was reduced, and the growth performance was also inhibited [9]. Ono et al. found that FSBM could increase the expression of related genes of the mTOR signaling pathway [33]. In addition, the PI3K/AKT/mTOR pathway was not only an important pathway for protein synthesis but also regulated fat metabolism [34]. This also explained that partial replacement of FM with SBM in this study could significantly reduce the contents of muscle crude protein, muscle crude fat, serum protein (TP and ALB), and T-CHO of juvenile coho salmon, while partial replacement of FM with FSBM tended to increase the above indicators of juvenile coho salmon.

In this study, we found that partial replacement of FM with SBM significantly decreased the digestive enzyme activity of juvenile coho salmon, and partial replacement of FM with FSBM significantly increased the digestive enzyme activity of juvenile coho salmon. We speculated that the contents of muscle crude protein, muscle crude fat, serum protein (TP and ALB), and T-CHO of juvenile coho salmon were not only related to the synthesis of the mTOR signaling pathway in vivo but also related to the digestion and utilization of proteins and lipids. First, SBM contained many anti-nutrient factors, which affected the palatability of feed and reduced the feed intake of SBM. FSBM could not only eliminate the influence of anti-nutrient factors but also improve the palatability of SBM and the utilization rate of protein and lipids [35]. At the same time, soybean peptides and

other substances produced by the fermentation process could also interact with digestive enzymes to improve the digestion and absorption of protein and lipids in feed [36]. Second, excessive SBM would affect intestinal health, cause inflammation of the small intestine epithelium, and affect the digestion and absorption of protein and lipids. FSBM could enhance the intestinal epithelial barrier and improve intestinal health [23,37]. Samad et al. also found that partially replacing FM with FSBM could reduce the feed conversion ratio and improve protein efficiency and digestive enzyme activities of spotted seabass [38]. Cunha et al. found that the partial replacement of FM by FSBM could improve the digestive enzyme activity of goldfish (*Carassius auratus*) [39].

Superoxide dismutase (SOD) and catalase (CAT) are the most important key enzymes in the antioxidant defense [40]. In this study, we found that partial replacement of FM with SBM significantly decreased the antioxidant capacity of juvenile coho salmon, and partial replacement of FM with FSBM significantly increased the antioxidant capacity of juvenile coho salmon. The reasons are supposed to be the following. First, the polyphenols contained in SBM were the main natural antioxidants in food. They were usually combined in the form of glycosides and could not be easily exploited [41]. Fermentation could not only produce more polypeptides with antioxidant activity but also facilitate the release of polyphenols in the glycosides [42]. Second, the immune-enhancing factors, bacteriostatic and isoflavones contained in FSBM, could remove the free radicals, inhibit lipid peroxidation, and enhance the antioxidant capacity [43]. Third, FSBM could stimulate the antioxidant system of the body and then inhibit the process of lipid oxidation and increase the content of T-CHO in serum [44]. Dan et al. also found that replacing FM with FSBM could significantly improve the antioxidant capacity of turbot (*Scophthalmus maximus*) [45]. Samad et al. fed spotted seabass the FSBM diet and found the level of T-AOC and the activities of SOD and CAT significantly increased [38].

Lysozyme (LYZ) is involved in the anti-infection and immune regulation of the body; it has the effects of antibacterial, anti-inflammatory, and antiviral; and it plays an important role in the body's defense against pathogen invasion [46]. Tumor necrosis factor $\alpha$ (TNF-$\alpha$), interleukin 1$\beta$ (IL-1$\beta$), and interleukin 6 (IL-6) are the cytokines in the innate immune system, and their levels are often used as indicators of the body's inflammatory response [47]. In this study, we found that partial replacement of FM with SBM significantly decreased the immune capacity of juvenile coho salmon, and partial replacement of FM with FSBM significantly increased the immune capacity of juvenile coho salmon. Similar studies have shown that FSBM could improve the immune capacity of Nile tilapia [26], turbot [48], and rainbow trout [49]. The reasons are supposed to be the following: First, the anti-nutritional factors in soybean meal can decrease the immune performance of aquatic animals and up-regulate the expression of genes related to inflammation [50]. Second, the plant polysaccharides, such as antimicrobial peptides and astragalus polysaccharides contained in leguminous plants, could be used as immune stimulants to improve the expression of immune genes in aquatic animals [51,52]. However, these beneficial components were not easy to digest and absorb by the body and were not enough to make up for the effects of anti-nutrient factors on the body [53]. Third, many bioactive peptides were produced after the fermentation of SBM, such as antimicrobial peptides, isoflavone glucoside, and lunasin [54,55]. They could not only eliminate the influence of anti-nutrient factors but also act as a regulatory compound, play the role of an immune stimulant, and reduce the expression of inflammatory genes [56,57]. Fourth, the cytoderm of fermented strain contained $\beta$-glucan and mannan oligosaccharides, which could play the role of immune stimulation and intestinal health promotion. The super adhesion of probiotics could regulate the balance of intestinal flora in fish and increase the number of immune cells in the mucosal layer, improve the level of blood antibodies, and regulate the inflammatory response of the body [58]. Yu et al. also found that feeding turbot with SBM instead of FM could increase the expression of cytokines and tumor necrosis factor [59]. Zhang et al. found that replacing FM with SBM could significantly increase the expression of pro-inflammatory factors in gentian grouper (*Epinephelus fuscoguttatus* $\times$ *Epinephelus Lanceolatus*) [60]. Li et al. found that 45% and 60%

FSBM feeding turbot instead of FM could decrease the expression of pro-inflammatory factors and tumor necrosis factors [61].

## 5. Conclusions

In summary, we found that replacing 10% FM protein with FSBM protein significantly improved the digestion, antioxidant, immune, and mTOR signaling pathway and had no negative effects on growth performance and serum biochemical indexes of juvenile coho salmon. Therefore, the FSBM protein could be used to replace part of the 10% FM protein in coho salmon feed.

**Author Contributions:** Conceptualization, Q.Z. and Y.L.; methodology, Q.Z., Q.Y., M.G., F.L., M.Q., Y.X., J.X. and Y.L.; investigation, Q.Z., Q.Y., M.G., F.L., M.Q., Y.X., J.X. and Y.L.; data curation, Q.Z., Q.Y., Y.L. and T.T.; writing original draft preparation, Q.Y.; writing—review and editing, Q.Z., Y.L. and T.T.; supervision, Y.L.; funding acquisition, Q.Z., Y.L. and T.T.; validation and visualization, Q.Z., Q.Y., M.G., F.L., M.Q., Y.X., J.X. and Y.L.; formal analysis, Q.Z., Q.Y. and Y.L.; project administration, Q.Z., Y.L. and T.T. All authors have read and agreed to the published version of the manuscript.

**Funding:** This research was funded by the Innovation Team Fund Project of Young Xiangsi Lake Scholars of Guangxi Minzu University (2018RSCXSHQN02), the Joint Funds of the National Natural Science Foundation of China (U20A2064), Shandong Provincial Key Research and Development Programs (2019JZZY020710), Innovation-driven Development Special Fund Project of Guangxi (AA17204044), and grants from the Scientific Research Foundation for the Introduced Talents of Guangxi Minzu University (2018KJQD14).

**Institutional Review Board Statement:** All animal experiments were conducted in accordance with the guidelines of Guangxi Minzu University, Nanning, China, and this research does not contain any studies with human participants (approval number: No. GXMZU 2021-006).

**Informed Consent Statement:** Not applicable.

**Data Availability Statement:** The original contributions presented in the study are included in the article, and further inquiries can be directed to the corresponding author(s).

**Conflicts of Interest:** The authors declare that they have no known competing financial interests or personal relationships that could have appeared to influence the work reported in this paper.

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
