# Peer review of "The Effects of Dietary Fermented Soybean Meal Supplementation on the Growth, Antioxidation, Immunity, and mTOR Signaling Pathway of Juvenile Coho Salmon (Oncorhynchus kisutch)"

_fishes, doi:10.3390/fishes8090448_

Round 1

Reviewer 1 Report (New Reviewer)

The manuscript entitled “Effects of dietary fermented soybean meal supplementation on growth, antioxidation, immunity, and mTOR signalling pathway of juvenile coho salmon (Oncorhynchus kisutch)” aims aims to investigate the effects of partial replacement of fish meal by soybean meal and/or fermented soybean meal on growth, serum biochemistry, digestion, antioxidation, immunity, and mTOR signaling pathway of O. Kisutch juveniles. The introduction is streamlined and clear, the experimental protocol well explained, the data obtained presented clearly and adequately discussed. The manuscript may be published with minor revisions as detailed below.

Introduction:

  • Line no 62-69: Write the importance of this study. Elaborate the research gaps.

Materials and methods:

·         Line no 77: the bacteria content was about 8 × 108 CFU/mL) ………. How the bacterial concentration was determined?

·         × 100% ……… Delete % (at the end) from the formula of WGR (%), SGR (%/day), PER (%), CF (%) and SR (g).

·         one-way analysis of variance was performed by IBM SPSS ……….. What is the name of test used in ANOVA? Mention it.   

Written very nicely

Author Response

Response to Reviewer 1 Comments (Round 1)

Point 1: Line no 62-69: Write the importance of this study.  Elaborate the research gaps.

Response 1: According to the Reviewer's suggestion, the research gaps have been added. “Coho salmon (Oncorhynchus kisutch) is one of the promising new Pacific salmon species in China due to its rich nutrition, delicious meat, and high economic value (Song et al., 2021). FM is considered by far the most advantageous protein source for coho salmon. However, there aren’t many reports on coho salmon fed FSBM instead of FM. The substitution of FM for FSBM was studied in coho salmon to integrate an assessment of its effects.”

Point 2: Line no 77: the bacteria content was about 8 × 108 CFU/mL) ……….  How the bacterial concentration was determined?

Response 2: The bacterial concentration is determined by colony formi ng units (cfu method). The specific operation is to take 1 mL of bacterial suspension and make 10-1, 10-2, 10-3, 10-4, 10-5, 10-6, 10-7, 10-8, 10-9 diluents by gradient dilution, and then take 1 mL from 10-6, 10-7, 10-8, 10-9 diluents and mix with Luria-Bertani medium in sterilized plate, cultured at 37 °C for 48 hours. Recorded the number of colonies formed in each plate and calculated the total number of viable bacteria contained in each mL of the original sample according to the dilution multiple. The whole process is in a sterile operating environment.

Point 3: × 100% ……… Delete % (at the end) from the formula of WGR (%), SGR (%/day), PER (%), CF (%) and SR (g).

Response 3: According to the Reviewer's suggestion, the % has been deleted.

Point 4: one-way analysis of variance was performed by IBM SPSS ………..  What is the name of test used in ANOVA?  Mention it.

Response 4: According to the Reviewer's suggestion, the name of test used in ANOVA has been added. “All the data were preliminarily processed by Microsoft Excel 2021 (Microsoft Corporation), and a one-way analysis of variance (ANOVA) was performed by IBM SPSS 21 (International Business Machines Corporation). The normality and homogeneity of variances among groups were tested. Tukey’s multiple range test was used to compare each other. When p < 0.05, it is significantly different.”

Once again, thank you very much for your comments and suggestion.

Reviewer 2 Report (New Reviewer)

The goal of the paper is to provide enough scientific evidence for the aquaculture industry to partially replace fish meal with fermented soybean meal as a sustainable alternative for the commercial diet of coho salmon (Oncorhynchus kisutch). The experimental design and chosen investigations are appropriate and the results are clearly presented in order to be considered a paper worthy to be published.

However, there are small issues that, if addressed, could improve the quality of the paper:

 The introduction section could be expanded to accurately describe the current state of knowledge: there are many published studies regarding the replacement of FM with FSBM with contradictory results (regarding the level of inclusion and results); different strategies to obtain FSBM should be mentioned underlining the similar studies' results and the gap in knowledge... (besides the addressed specie).

Some sentences could be reformulated for meaning clarity (phrases from lines 45-49, 173, 198, 291-194, 313-317, 322-325 ).

The common names of the species should be written similarly over the whole document, either with capitals or normal letters.

Line 321: erase ”of the mTOR 320 signaling pathway” from the end of the sentence. 

Line 336: it is a speculation or a hypothesis?

Line 339-241: What is the direct connection between palatability and digestive enzyme secretion?  The phrase is not linked correctly to the main subject of the paragraph and should be rather connected with a sentence from lines 348-350.

Line 366: Avoid formulation like ”Similar studies have shown that Dan et al. (2022)....” First: there is only one study (not studies) cited; second: the reference is better to be at the end of the sentence. Moreover, the journal Fishes has rules regarding the reference style, rules that authors should follow ”In the text, reference numbers should be placed in square brackets [ ], and placed before the punctuation; for example [1], [1–3] or [1,3]. For embedded citations in the text with pagination, use both parentheses and brackets to indicate the reference number and page numbers; for example [5] (p. 10). or [6] (pp. 101–105).

When a list of reasons follows, try to keep the explanations short and separate them with ;  otherwise, the reader cannot follow the reasoning of the authors and the argument is lost (see lines 357-366, 380 -399). 

The English could be improved.

Author Response

Response to Reviewer 2 Comments (Round 1)

Point 1: The introduction section could be expanded to accurately describe the current state of knowledge: there are many published studies regarding the replacement of FM with FSBM with contradictory results (regarding the level of inclusion and results);   different strategies to obtain FSBM should be mentioned underlining the similar studies' results and the gap in knowledge...   (besides the addressed specie).

Response 1: According to the Reviewer's suggestion, the introduction section has been expanded. “There are many published studies regarding the replacement of FM with FSBM with contradictory results. The study showed that using FSBM instead of FM to feed spotted seabass (Lateolabrax maculatus) has high growth performance and immune ability and the optimum replacement level of FM with the FSBM was estimated to be in the range of 26.9–37.1%. However, up to 30% FM in the diets of juvenile yellowfin (Acanthopagrus latus) could be replaced by FSBM, and the growth performance and whole-body proximate compositions of fish were not affected by dietary FSBM level. Therefore, considering the different response strategies of different species to FSBM, and few research on the effect of FSBM instead of FM on salmon, finding suitable levels of FSBM and probiotics is very important for the development of salmon industry.”

Point 2: Some sentences could be reformulated for meaning clarity (phrases from lines 45-49, 173, 198, 291-194, 313-317, 322-325 ).

Response 2: According to the Reviewer's suggestion, the sentences have been reformulated for meaning clarity.

Point 3: The common names of the species should be written similarly over the whole document, either with capitals or normal letters.

Response 3: According to the Reviewer's suggestion, the common names of the species have been written with normal letters.

Point 4: Line 321: erase ”of the mTOR 320 signaling pathway” from the end of the sentence.

Response 4: According to the Reviewer's suggestion, “of the mTOR signaling pathway” has been deleted.

Point 5: Line 336: it is a speculation or a hypothesis?

Response 5: It is a speculation. What I want to explain is that the previous changes in serum biochemical indicators may also be related to digestive enzyme activity.

Point 6: Line 339-241: What is the direct connection between palatability and digestive enzyme secretion?      The phrase is not linked correctly to the main subject of the paragraph and should be rather connected with a sentence from lines 348-350.

Response 6: According to the Reviewer's suggestion, this paragraph has been rewrited.

Point 7: Line 366: Avoid formulation like ”Similar studies have shown that Dan et al. (2022)....”     First: there is only one study (not studies) cited;     second: the reference is better to be at the end of the sentence.     Moreover, the journal Fishes has rules regarding the reference style, rules that authors should follow ”In the text, reference numbers should be placed in square brackets [ ], and placed before the punctuation;     for example [1], [1–3] or [1,3].     For embedded citations in the text with pagination, use both parentheses and brackets to indicate the reference number and page numbers;     for example [5] (p. 10).     or [6] (pp. 101–105).”

Response 7: According to the Reviewer's suggestion, the reference format has been rewrited.

Point 8: When a list of reasons follows, try to keep the explanations short and separate them with ;      otherwise, the reader cannot follow the reasoning of the authors and the argument is lost (see lines 357-366, 380 -399).

Response 8: According to the Reviewer's suggestion, this paragraph has been rewrited.

Once again, thank you very much for your comments and suggestion.

This manuscript is a resubmission of an earlier submission. The following is a list of the peer review reports and author responses from that submission.

Round 1

Reviewer 1 Report

This study that focuses on the genomic aspect of fish nutrition has very good scientific merit on the surface but has deeper limitations of many kinds. The technical work is obviously originating from a highly competent grouping in China with a clear grasp of molecular biology and genetics. The interdisciplinary nature of the work to connect nutrigenomics with aquaculture nutrition is interesting and can lead to new avenues. However, some aspects of basic aquaculture comprehension relating to the applied requirements of feed ingredient use and definition appears to be weaker.   The description of each feed ingredient must be given in full and its sources and type of processing transparent. Describing materials as chicken powder, flour etc is very unacceptable in animal feed experiments.  They say that Soybean meal 35 (SM) is a potential protein substitute for feed. We know this statement and soybean has been used for a very long time, so it’s not a potential but factual aaaaand used.  There are many more ingredients that are now being evaluated to even replace soybean as it is seen and appreciated to be non-sustainable as well as fishmeal. (Oncorhynchus kisutch) has become the most promising new breed of salmon trout in China (it is NOT a salmon trout but a Pacific salmon that is a relative of rainbow trout. This must be addressed and makes it to me clear that you do not full appreciate aquaculture on a global level. Pacific salmon are reared but not anywhere near the levels of Atlantic salmon. you must mention this!

In fact, the introduction does not give any rationale for measuring various enzymes and other metabolites etc. you should describe in the introduction the basis for specific selections and how they may relate to provide us with added information that we can actually use to improve our understanding of the mechanics. I am confident that you fully understand the molecular theory and use of gene expression data, but the industry may not find this deep dive into genomics so attractive into any application without good reasons. The diet formulations being used currently are highly variable and this study is rather specific around soybean meal making many assumptions. Soybean meal also is very variable in quality and you are not able here to characterise its nutritional value and level of antinutrients. it is all rather vague.

You say much about antinutritional factors, but you did not assay for them. In fact, trypsin inhibitor proteins are the main ANF in soya, but your data shows little evidence of any inhibition at the inclusion levels tested. How is this factored in your discussion.

Trypsin (U/mgprot) 2168.31±17.36b 1569.23±42.29c 2115.97±88.12b 2605.51±9.57

The manuscript has a disjointed discussion and fails on lack of evidence behind conjecture trying to explain the results obtained. This is the danger of molecular genetics branching into aquaculture just because one has the ability to make advanced methods in molecular genomics etc. I do think you must really provide much more information of the quality of the major ingredients both soy and feremented soy to link with your enzyme assays and metabolite measurements and gene experssion etc. The immunology aspects is much better and clearer regarding effects. I think you are trying to overstretch the science and interpretations in too many ways and this fails to project a strong message and useful conclusions. 

The English is OK but there is much more scope to stramline the grammar in places. Some sentences need to be more flowing.

Author Response

Point 1: Describing materials as chicken powder, flour etc is very unacceptable in animal feed experiments.

Response 1: According to the Reviewer's suggestion, the “chicken powder” has been changed to “poultry meal”. The “flour” has been changed to “wheat middling”.

Point 2: They say that Soybean meal (SM) is a potential protein substitute for feed. We know this statement and soybean has been used for a very long time, so it’s not a potential but factual and used.       There are many more ingredients that are now being evaluated to even replace soybean as it is seen and appreciated to be non-sustainable as well as fishmeal.

Response 2: According to the Reviewer's suggestion, this sentence has been rewrited. “Soybean meal (SBM) has been widely used as a feed protein for many years.”

Point 3: (Oncorhynchus kisutch) has become the most promising new breed of salmon trout in China (it is NOT a salmon trout but a Pacific salmon that is a relative of rainbow trout. This must be addressed and makes it to me clear that you do not full appreciate aquaculture on a global level. Pacific salmon are reared but not anywhere near the levels of Atlantic salmon. you must mention this!

Response 3: According to the Reviewer's suggestion, this sentence has been rewrited. “Coho salmon (Oncorhynchus kisutch) is one of the promising new Pacific salmon species in China.”

Point 4: You say much about antinutritional factors, but you did not assay for them. In fact, trypsin inhibitor proteins are the main ANF in soya, but your data shows little evidence of any inhibition at the inclusion levels tested. How is this factored in your discussion.

Response 4: According to the Reviewer's suggestion, the nutritional composition and anti-nutritional factor content of soybean meal and fermented soybean meal have been added, shown in Table 1.

In this study, the trypsin inhibitor proteins are the main ANF of soybean meal, but soybean meal replaces some of the fish meal, but not all of it, so the trypsin activity was only partially inhibited, not completely deactivated. Compared with other experimental groups, the trypsin activity in G1 group was significantly inhibited.

Once again, thank you very much for your comments and suggestion.

Reviewer 2 Report

This paper is well written, Introduction, methods an results seems fine, also the conclusions are supported by the results.

However, I am concerned about the data showed. How do the authors get the weights of the fish to be exactly the same at the start of the experiment?

I am also concerned about the standard deviation shown in the expression data, I have never seen such a low standard deviation before. Are you sure you are plotting SD and not SEM, in Prism you can easily cofuse between one and the other. Please make sure your calculations are correct. I think you should also show the individual data points on your graphs to clarify this issue.

In addition, the authors should add the data related to the ANOVA analysis, as the p-values and F-values would be very useful to understand the results. 

Author Response

Response to Reviewer 2 Comments

Point 1: However, I am concerned about the data showed. How do the authors get the weights of the fish to be exactly the same at the start of the experiment?

Response 1: In this study, five hundred juvenile coho salmon were purchased from the rainbow trout breeding farm. 500 juvenile coho salmon hatched at the same time, and the juvenile coho salmon were all about six months old. Then, 240 juvenile coho salmon were used for this experiment. In the process of selecting fish, we try to choose the same size, healthy and disease-free fish for the experiment. 240 juvenile coho salmon were randomly divided into 4 groups in triplicate. 3 fish were randomly selected from each experimental net cage to measure weight and take the average value. So the weights of the fish to be exactly the same at the start of the experiment.

Point 2: I am also concerned about the standard deviation shown in the expression data, I have never seen such a low standard deviation before. Are you sure you are plotting SD and not SEM, in Prism you can easily cofuse between one and the other. Please make sure your calculations are correct.

Response 2: According to the Reviewer's suggestion, the experimental data have been recalculated. Results were expressed as Mean ± standard deviation (Mean ± SD).

Point 3: In addition, the authors should add the data related to the ANOVA analysis, as the p-values and F-values would be very useful to understand the results.

Response 3: According to the Reviewer's suggestion, the P-values and F-values have been added.

Once again, thank you very much for your comments and suggestion.

Round 2

Reviewer 1 Report

Thank you in paying attention to improve the manuscript according to my suggestions in several areas. It is now much better and largely addresses my concerns. I am happier with it now as contributing good knowledge and contributing to the topic of fish nutrition and alternative proteins with robust data to link with new perspectives for rearing Pacific salmon species. This includes novel emphasis on their metabolic and enzyme status which is good work.

Author Response

Dear Reviewer, Thank you very much for your comments and suggestion.

Reviewer 2 Report

The authors have not made the necessary changes to improve the manuscript as expected. Furthermore, the statistics applied should be revised as they do not correspond to the data provided.

Author Response

Response to Reviewer 2 Comments

Point 1: The authors have not made the necessary changes to improve the manuscript as expected.  Furthermore, the statistics applied should be revised as they do not correspond to the data provided.

Response 1: According to the Reviewer's suggestion, all data have been recalibrated and processed using SPSS software. The results section has been rewritten based on the data processed. I'm sorry that I didn't understand your first suggestion on the charts section last time. According to your first review, the charts section has been completely revised.

Once again, thank you very much for your comments and suggestion.

Round 3

Reviewer 2 Report

Unfortunately, the results presented in this paper do not allow me to accept it.

The authors should explain how it is possible that the initial weight values of the fish have been modified from the first version of the MS to the last one. The same happens with other parameters displayes, the values of WGR that changes from 237.78 to 336.52, in G0, 273.41 to 173.41 in G1... same happened with HSI data, F values vary from 2.8 to 13.2. p-value from 0.1 to 0.002... among others.

The authors still do not provide the individualised data as asked, neither raw data to justify these changes

There are many doubts about the data presented, the authors must make a great effort to repay the trust with this work.

Unfortunately the data presented in this study lacks reliability and I cannot do anything but reject it.

I hope that the authors will make an effort to be more transparent in the future and present work that meets scientific standards, otherwise we are doing a disservice to the global scientific community.